



# Deep oceanic submarine fieldwork with undergraduate students, an exceptional immersive experience (Minerve software)

Marianne Métois[1], Jean-Emmanuel Martelat[1], Jérémy Billant[2], Muriel Andreani[1], Javier Escartín[3], Frédérique Leclerc[2], and ICAP [4]

[1]Laboratoire de géologie de Lyon, Université de Lyon, ENS de Lyon, CNRS, UMR5276 LGL-TPE, 69622 Villeurbanne, France
[2]Université Côte d'Azur, CNRS, Observatoire de la Côte d'Azur, IRD, Géoazur, 250 rue Albert Einstein, Sophia Antipolis 06560 Valbonne, France.
[3]Laboratoire de Géologie, UMR 8538, Ecole Normale Supérieure, PSL Research University, CNRS, Paris, France
[4]iCAP (Innovation, Conception, Accompagnement pour la Pédagogie), University Lyon 1 43 Bd du 11 novembre, 69622 Villeurbanne, France

**Correspondence:** marianne.metois@univ-lyon1.fr

**Abstract.** We present the content and scripting of an active tectonic lab-session conceived for third year undergraduate students studying Earth Sciences at *Observatoire des Sciences de l'Univers* of Lyon. This session is based on a research project conducted on the submarine Roseau active fault in Lesser Antilles. The fault morphology is particularly interesting to map as this structure in the deep ocean is preserved from weathering. Thus high resolution models computed from Remotely Operated

Vehicle videos (ROV) provide exceptional educational material to link fault morphology and coseismic displacement. This class, composed of mapping exercises on GIS and virtual fieldwork, aims at providing basic understanding of active tectonics, and in particular active fault morphology. The work has been conducted either in a full remote configuration via 3D online models or in virtual reality (VR) in a dedicated room using the Minerve software. During the VR sessions, students were either alone in the virtual environment or participated as a full group, including the teacher (physically in the classroom or remotely,

from another location), which is to our knowledge one of the first attempts of this kind in France. We discuss on the efficiency of virtual fieldwork using VR based on feedback from teachers and students, and we conclude that VR is a promising tool to learn observational skills, subject to certain improvements which should be possible in the years to come.

## 1 Introduction

Metropolitan France is a relatively quiet country in terms of seismic hazard (e.g. Duverger et al., 2021). Some Mw6+ destructive

earthquakes were registered in the past (such as the Lambesc, 1909 Provence earthquake (e.g. Baroux et al., 2003)), and the 2019 Mw5 Le Teil earthquake that reactivated a branch of the Cevennes fault system recall that seismic hazard is not zero (Ritz et al., 2020; Cornou et al., 2021). However, because Metropolitan France is a very slowly deforming region where deformation is diffuse (Masson et al., 2019), the morphological signature associated with potentially active faults is often subtle. Taking undergraduate students to the field to observe an active fault with clear morphological trace requires therefore to go abroad to

more tectonically active areas (e.g., Italy, Greece), thus involving rather long and expensive field-work sessions.



However, fieldwork is essential in Earth Sciences learning. The observation of geological objects in situ, their 3D and 2D representation, are key to decipher their nature and geological history. While often considered as attractive for students that appreciate facing the subjects of study, fieldwork may be unfeasible (e.g., dangerous, remote or submarine places). We may require instead virtual imaging to make a proper fieldwork-like analyses. The development of 3D visualisation and virtual

reality immersion in Geoscience offers an alternative path that starts to be explored and developed (e.g. Jitmahantakul and Chenrai, 2019; Mead et al., 2019; Janiszewski et al., 2020; Klippel et al., 2019).

In Lyon, the ICAP service (*Innovation, Conception et Accompagnement pour la Pédagogie*) from Université Lyon 1 opened a dedicated room for virtual reality teaching during spring 2020, equipped with 10 Oculus Rift S headsets connected to desktop computers, an interactive white board and collaborative facilities, that include multiple screen sharing (Mersive Solstice

system, see https://virtuallab.univ-lyon1.fr/). Simultaneously, a team of researchers involved in projects aiming at understanding the active tectonics of the French Lesser Antilles developed an interactive software (Minerve, see Billant et al., 2019) to work jointly on very high resolution (~1m to 10cm) DEM and DOM (Digital Elevation and Digital Outcrop Models) of the submarine active normal Roseau fault scarp that was reactivated during the 2004 Les Saintes earthquake (Mw 6.3) (Escartín et al., 2016).

We therefore had a unique opportunity to bring our students to the field, at 1200m below sea level, through virtual reality, and we describe this experiment in this paper.

## 2  Digital Outcrop Models visualization : from research to teaching

During the 2013 ODEMAR and 2017 SUBSAINTES cruises of the Flotte Océanographique Française (Escartín and Andreani, 2013; Escartín et al., 2017), the Roseau fault, that has a maximum vertical relief of 200 m, was imaged optically with the

Remotely Operated Vehicle (ROV) VICTOR, in addition to AUV microbathymetric surveys (Escartín et al., 2016; Istenič et al., 2020; Hughes et al., 2021). This deep-sea vehicle acquired high resolution videos at the base of the fault scarp, and along vertical transects of the fault plane, in order to study the 2004 earthquake submarine rupture. Georeferenced and scaled DOMs were calculated by applying and developing Structure from Motion Techniques on the videos (Istenič et al., 2020). In addition to this ROV survey, the Autonomous Underwater Vehicles AsterX (Ifremer, France) and Abyss (GEOMAR, Germany)

acquired near-bottom high resolution bathymetric data allowing to generate 1m-resolution DEMs. Both the optical and the high-resolution acoustic data were used to describe and quantify the coseismic displacement of the 2004 event, and to better understand and quantify the submarine landscape evolution processes that shape the submarine fault morphology (Escartín et al., 2016; Hughes et al., 2021).

However, existing software allowing to work simultaneously on DEMs and DOMs are scarce, or ill-suited as they often lack

georeferencing (e.g., Meshlab). Several of these systems also allow the user to interact with the 3D data on screens only, and not in a VR environment (e.g., Matlab, QPS Fledermaus, Matisse (Arnaubec et al., 2021)), among other limitations. Therefore, in order to precisely analyze and inspect interactively structures from 3D models, and in this case the fine scarp topography and texture of a fault rupture underwater, the Minerve Virtual Reality software was developed (Billant et al., 2019). This



development is intended to be used as a quantifying tool, and provided as a free and open source software. In Minerve, the

user can move freely in a georeferenced space at 1:1 scale. The tools allow measuring strike, dip, rake, and distances, and mapping geological features that can be exported in GIS-like format for further work. Lastly, several users can remotely meet and collaborate in the same VR environment, allowing team work, or facilitating training and teaching. The possibilities of interaction with the 3D models offered by Minerve are key skills that students in Geosciences should master at the end of the License (3rd year undergraduate students).

Minerve software was used to perform a paleo-seismological study of the Roseau fault outcrops (Billant et al., 2018). Although the fault was imaged in 2013, 13 years after the earthquake, the morphology and visual texture of the coseismic markers are astonishingly well preserved since at this oceanic depth (∼1200m) weathering and sedimentation rates are very slow (Escartín et al., 2016), and constitute text-books normal fault outcrops. Such markers are much more ephemeral in subaerial environments. For instance, light coseismic ribbons at the base of darker cumulative scarps are usually fainted rapidly inland

while the color change is still clearly visible along the Roseau fault. Several markers of pre-seismic seafloor levels imprinted on the fault mirror, such as thin lines of sediment stick on the fault mirror, are also preserved at different elevation, making the Roseau fault outcrops unique to discuss seismic cycle as well as tectonic and submarine surface processes interaction, especially with undergraduate students.

## 3 Experience design: an introduction to active tectonics

The lab session presented in this study has been built for third year undergraduate geosciences students (L3), and has been tested by students following the General Geology undergraduate program in Université Lyon 1 and École Normale Supérieure de Lyon. It aims at providing them with basic understanding of active tectonics in the frame of a more general course on "Structural Geology and Tectonics". The lab session is associated with a 3 hours lecture on seismic cycle, scaling laws, and morphology of active faults. At the end of the course unit, the students can be evaluated on their ability to (i) understand a

tectonic context based on fault maps, focal mechanisms, earthquake catalogues and DEM, (ii) estimate a recurrence time for a given fault based on historical earthquake time-line and tectonic strain rate, (iii) estimate the maximum magnitude and type of earthquake that could generate a fault based on the scarp morphology, the length of the fault, and the standard scaling laws.

The teaching sequence, adapted for groups of up to 12 students, starts by mapping exercises and understanding of the tectonic context using vector and raster images gathered on a QGIS project. After identifying potential candidate faults for the

2004 Les Saintes rupture at a large scale using a 10m resolution DEM covering the whole fault system (Deplus and Feuillet, 2010; Leclerc et al., 2016), students ideally switch to VR immersion for a finer analysis using 1m resolution DEM and the cm resolved DOMs. Students' feedback was collected immediately after the lab session via an online inquiry, and its analysis will help us in the future to improve both the virtual fieldwork for the students, and the Minerve software itself (ergonomics, functionalities, tools).



Because of the Covid-19 sanitary restrictions, we adapted the first session conducted in spring 2020 during full lockdown in France in a 100% virtual lab-session. In spring 2021 during partial lockdown, we conducted the sessions at the university with 4 reduced groups composed of 4 to 6 students each, during short 2-hour sessions, including 1h dedicated to VR fieldwork.

## 3.1 Using GIS tools to analyze the tectonic context and geomorphology

Geographical Information Systems (GIS) are now used in a very wide variety of fields, including Earth Sciences. This lab-
session is the first contact for the third-year undergraduate students with GIS software and aims at introducing basic digital mapping tools and familiarizing the students with the tectonic context. The QGIS software (QGIS Development Team, 2021) was used during our courses.

In addition to standard documents extracted from the scientific bibliography (context figures extracted from (Feuillet et al., 2002; Leclerc et al., 2016) and USGS description of the 2004 Les Saintes mainshock including focal mechanism and inten-
sity map, see https://earthquake.usgs.gov/earthquakes/eventpage/usp000d8w3/executive), students are provided with a zip file including a QGIS project file and associated layers (provided as supplementary information). The project file can be opened directly by QGIS (version>2.7) without the need for manual tuning of projection parameters and symbology or loading of additional layers (operations that students do not yet know how to perform). The project includes vector and raster layers that are listed in table 1.

| Name | Type | Description |
|---|---|---|
| Seismic catalog | vector, points | 2004-01 to 2016-12 regional seismicity including the Les Saintes mainshock and aftershock sequence (0<Mw<7.5). Extracted from IPGP's seismological and volcanological observatories datasets, http://volobsis.ipgp.fr (Bazin et al., 2010) |
| Plate boundaries | vector, lines | Plate boundaries location and style (Bird, 2003, and https://github.com/fraxen/tectonicplates) |
| ROV Path | vector, points | Position of the ROV over time |
| DEM 10m | raster | bathymetry at 10 m resolution from several cruise surveys; in this article we provide instead the bathymetry grid from (Deplus and Feuillet, 2021), which has a 25m resolution, and was also acquired during the Bathysaintes cruise (Leclerc et al., 2016; Deplus and Feuillet, 2010) |
| DEM shadow | raster | Shadow derived from the DEM |
| DEM texture | raster | Texture derived from the DEM |
| Open Street Map | XYZ tile | Standard background OSM provided with QGIS |

**Table 1.** Description of the layers imported in the QGIS project (see supplementary material)

As a first step, students are asked to describe the overall tectonic context (plates in contact, type of plate boundaries, expected long-term motions) and to discuss the occurrence of a major normal-fault earthquake in a context of subduction, based on the

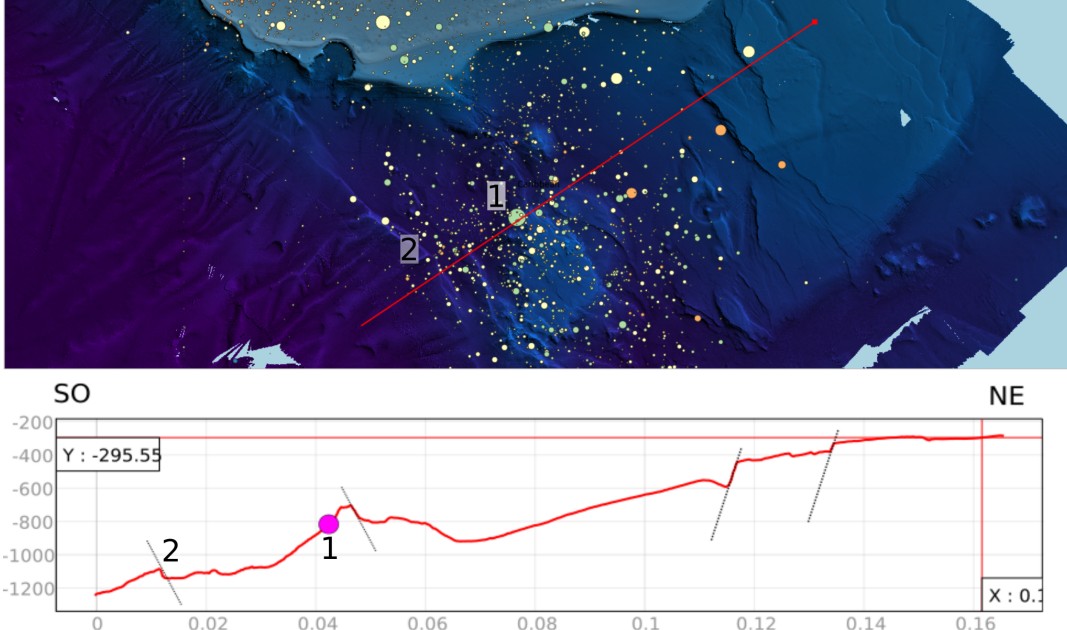

**Figure 1.** Snapshot of the QGIS desktop showing an example of interpretation by a student of a bathymetric profile (bottom panel) crossing the normal fault system perpendicularly (red line on the upper panel map) using the Terrain Profile tool. The student has represented supposed normal fault planes with dashed line and the mainshock epicenter with purple dot (1). The Roseau fault is indicated both in map and profile views (2).

standard documents provided. They are then guided through the QGIS project to have a closer look to the seismicity catalog, and to identify the 2004 sequence via simple requests on the attribute table (sorting, request on attribute value, etc).

As a second step, students work on the interpretation of the shaded and textured 10m local DEM. They explore the DEM via
the "identify features" tool to get absolute bathymetry value at each pixel, and the "Terrain Profile" additional plugin (https:// github.com/PANOimagen/profiletool) that allows on-the-fly drawing of topographic profiles (see figure 1). Bathymetric profiles perpendicular to the Roseau fault system can be easily interpreted as representative of an active graben since cumulative fault scarps are very well preserved. Students are asked to use drawing tools (line, polygons) to provide a simplified structural map of the area including active faults, volcanoes and reef plateau (figure 2). Doing so, they have to wonder about the relationship that exist between these structures and propose a chronology for their setting. They can use the layout manager to finalize their
work in the form of a synthetic and commented map.

Once this mapping exercise is over, students are asked to use their morphological observations to propose a fault as the best candidate for hosting the 2004 Les Saintes event. To do so, they have to remember that (i) such a large earthquake (Mw 6.3) requires a ∼15km long fault segment to rupture if agreeing with standard scaling laws, (ii) the epicentral location is often shifted from the surface fault trace due to fault dip and depth of the rupture (here 12km) and therefore (iii) have to take into
account the standard dip for normal faults that is in theory around 60-65° (Anderson, 1951; Olive and Behn, 2014). All together,





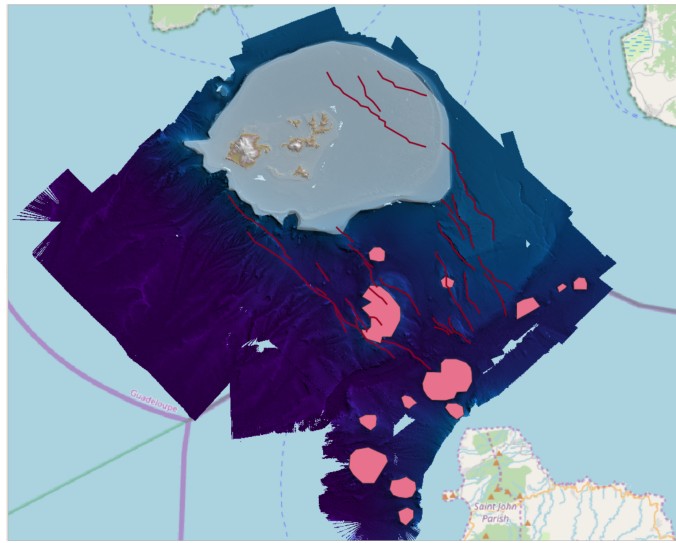

**Figure 2.** Snapshot of the QGIS desktop showing an example of expected first-order interpretation of the 10m DEM using QGIS drawing tools. The Les Saintes reef plateau is contoured in transparent white, submarine volcanoes in pink, and active faults scarp in red. Erosional features like canyons could also be mapped since they are often detected by students and may be misidentified as tectonic scarps.

these parameters should lead students to conclude that detailed submarine fieldwork should be conducted on the Roseau fault to look for a fresh scarp.

## 3.2   On the field : virtual reality and 3D models

Before going on the virtual field, students are provided with a excerpt of the raw video taken by the ROV during the SUB-SAINTES cruise (ROV Dive 563), from which the detailed DEM is built (see https://www.youtube.com/watch?v=TV3TUeRfxoc). This video enables to discuss the technical difficulties that arise when doing submarine exploration. In addition to the absence of a landscape view that provides a general reference to the user, these data illustrates several limiting factors: restricted field of view, poor visibility due to sediment particles, artificial lighting that is thus distance-dependant, no direct scaling, difficulties in

orientation due to camera and vehicle motion, no GPS-positioning etc (Istenič et al., 2020). At the same time, some advantages of submarine exploration and field work can also be pointed out, such as the very limited erosion rate and good preservation of structures, and the accessibility to the outcrop as is not covered by vegetation, as on-land.

With the virtual fieldwork, the students' goals are: describing and identifying the morphology associated with the active fault, measuring the last coseismic displacement on fault scarps and estimate the moment magnitude of the last earthquake, mapping

the fault, and understanding the erosive and sedimentary processes interacting with tectonics (dejection cones, roughness of the scarp, etc). Students ultimately propose scenarios of fault behavior during the seismic cycle.

We have tested three different teaching strategies pictured on figure 3. In the first strategy, called "100% remote", students work from home on their own laptop during the entire session. They are connected to a dedicated voice channel on a Discord





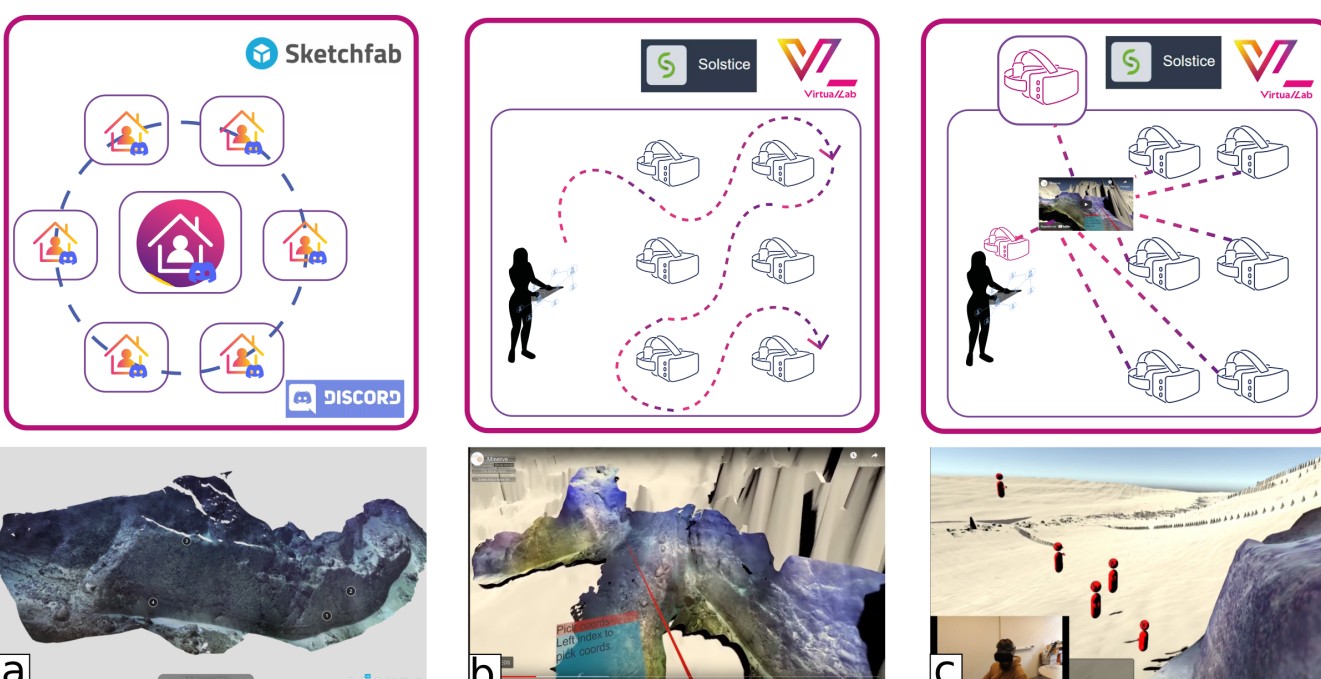

**Figure 3.** Teaching strategies tested for virtual fieldwork sessions (upper panel) together with a view of the studied outcrop (lower panel). a- "100% remote" strategy based on the Discord app and the use of a 3D model hosted on the Sketchfab platform; b-"Alone in the field" or "single-user" strategy taking place in the Virtual Lab with the teacher (dark silhouette) remaining outside the VR environment and guiding the students via the Solstice system; c-"Together in the field" or "multi-user" strategy in the virtual lab with two teachers connected to the VR environment.

server hosting the other students and the teachers. The fieldwork is conducted based on a short portion of the 3D DEM/DOM
(centimetric resolution) of the Roseau scarp loaded in degraded resolution (50 Mb in total size, presenting a quarter of the complete model at less than a quarter of its initial resolution unlike explored in other strategies) on a Sketchfab account (see https://sketchfab.com/3d-models/la-rf-fpa-85002c5cd5f54a8fbeb736576b7d9e91). Few annotations are added to the model in order to discuss some specific points with the students and provide a rough estimate of the outcrop's scale.

In the second strategy, called "Alone in the field" or"single-user mode", the students are present on the same classroom but
separated in two different groups and sessions. They are alone in the virtual field and guided by a teacher from outside the VR environment in the Univ. Lyon 1 virtual lab (see figures 3-b and 4). The teacher can follow the displacements of the students and share their view via multiple screen-sharing projected on the digital board (see annotation 1 in figure 4).

In the third strategy, called "Together in the field" or "multi-user mode", the students are separated in two small groups (6 students/group) and are all together in a single VR environment with the instructor (J. Billant), teaching remotely from Nice
town (see figures 3-c and 5). A "local" teacher is also present in the virtual lab and in the VR environment depending on the student needs.





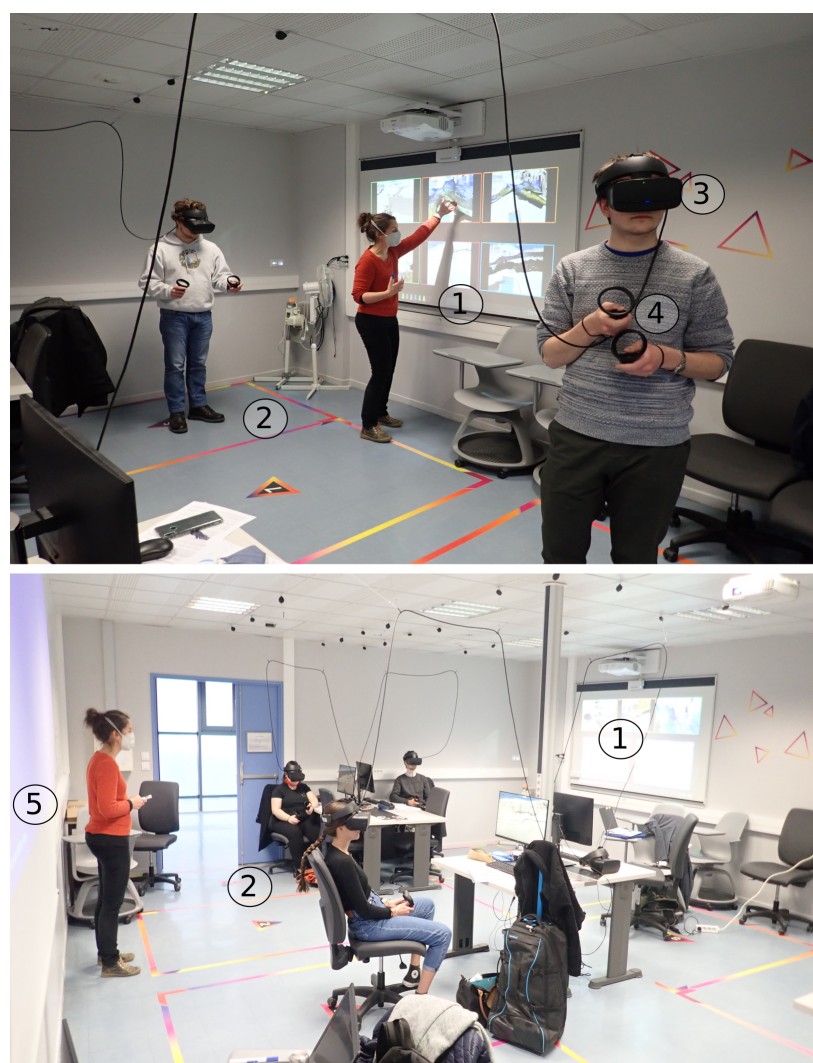

**Figure 4.** Views of virtual lab room in the "Alone in the field" or "single-user" mode in which teacher (in orange) is outside the virtual environment. 1- Solstice collaborative screen sharing system, 2- virtual box where limited real displacements of the user are allowed, 3- Oculus Rift S headset, 4- controlers, 5- projection screen for teacher computer. It is to note that some students prefer to operate in VR while seated, while others find it more natural to be standing or moving in the limit of their virtual box (2).



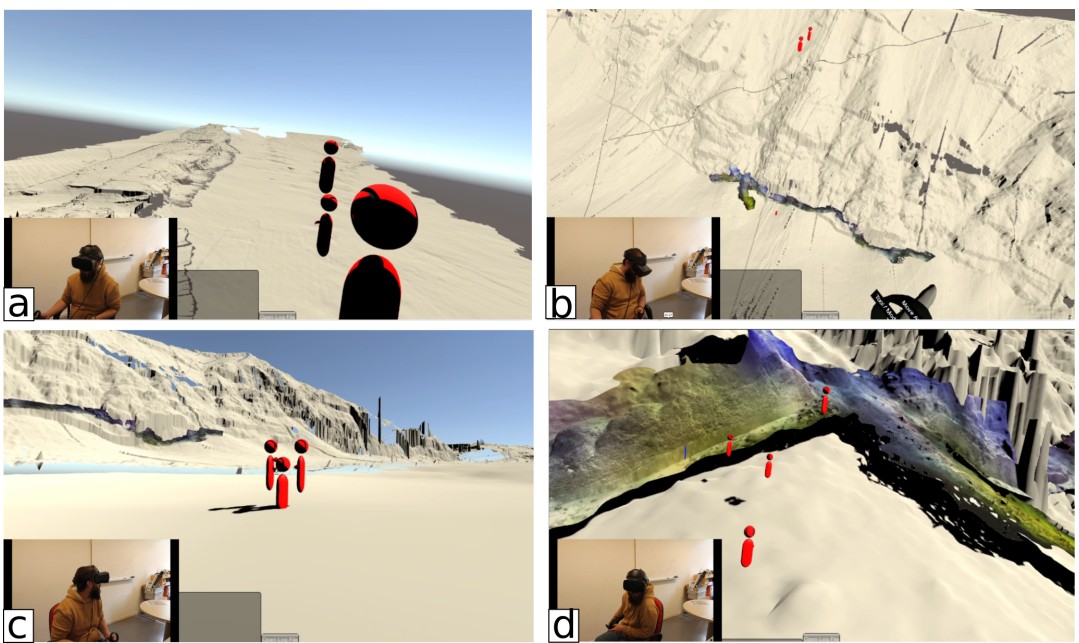

**Figure 5.** Views of the virtual environment in the "Together in the field" or "multi-user" mode from the remote teacher perspective (insert in lower left). a- Distant view of the 1m resolution DEM (beige), reconstruction artifacts are greyish zones. The teacher presents the entire tectonic structure that is 200m high. The colored portion of the scarp is the centimetric resolution DOM of the fresh scarp (6m height and 220m long). Students are red avatars. b and c- The student discuss the structure and move closer. d- Field mapping of details observable in the HR model at 1:1 scale.

For the two strategies involving virtual reality, we dedicated limited time (5-10 minutes) to take control of the virtual tools and motion modes in the virtual environment (rotation, translation at different speeds, teleportation, flying mode). We take advantage of this period during which the students are still not autonomous on the field to make them look at the 3D model

in its entirety, i.e. from above using the flying mode (figure 5a-b). This first step allows the students to orient themselves, to measure the overall fault azimuth using the wrist compass and to appreciate the total height of the cumulative scarp (i.e. ~200m), and length of the mapped model. Particular care is taken to help the students spotting the artifacts of the rough large scale DEM (1m resolution) in order to avoid misinterpretations. It gives the opportunity to discuss the technical difficulties in building such a DEM from onsite measurements (e.g. Debese, 2013).

In a second step, the students are free to explore the model and are encouraged to have a closer look to the very high resolution part of the DOM (1-5cm) that covers a 220m long and 6m high portion of the scarp (figure 5b-c). They have to propose a precise mapping of the fault (draw line tool), an estimate of the slip associated with the most recent 2004 earthquake (distance measurement), and to measure the strike and dip of the fault plane and rake of the striae (compass tool). All these measurements can be saved in comma separated values format as georeferenced features with attributes and can therefore be

loaded in the student's QGIS project built in a first step (see section 3.1) to complete their analysis.



Finally, the teacher guides the students through the detailed geological interpretation of the outcrop in order to detect changes in scarp roughness or color, traces of old sediments on the fault scarp, erosive steps, etc that should help the students in first producing an annotated synthetic sketch of the outcrop, and second discuss the regularity of the seismic cycle over this major fault. For a complete and detailed analysis of the outcrop, see Escartín et al. (2016); Hughes et al. (2021); Billant et al. (2018).

Not all of these objectives could be reached using the "100% remote" strategy that limits the students analysis of the outcrop to a qualitative description of a very limited part of the 3D model. Therefore, this strategy is rather a virtual tour than an actual virtual fieldwork. In the following section we analyse the drawbacks and advantages of all three educational attempts.

## 4 Discussion

### 4.1 Teaching strategies: comparison from teacher point of view

The "100% remote" strategy is obviously the least optimal, both regarding GIS study of the tectonic context and virtual field work since it is the least close to the real fieldwork. All the students were able to install QGIS on their personal laptop and were able to handle the layers of the project, at least for visualisation purposes. Most of them were successful in creating some bathymetric profiles and their own vector layers and sent us screenshots (see figure 1). However, remote debugging is difficult and many students did not participate to the oral discussion, that makes it difficult to conclude on their real understanding of

the concepts. Lab sessions in-person are more efficient to help the students both with technical issues and for guiding in the interpretation of the DEM. For this step and for small groups of students (6 students), the projection of all the active screens on the wall provided in the Lyon 1 Virtual Lab is a real advantage: it allows the teacher and the students to share their screen and easily discuss some features that may be difficult to locate or describe otherwise.

The virtual fieldwork via 3D online models on the screen had suffered mainly from the emergency context of 2020 lockdown

and the relative lack of preparedness of our team in using 3D online platforms such as Sketchfab or V3Geo (Buckley et al., 2021). Our attempt therefore suffered from several technical limitations that will be overcome in the future, but also from inherent issues due to full remote teaching. The use of a free Sketchfab account limited in features imposed a reduction of the size of the model, therefore preventing a detailed analysis of some fine structures and forcing the students to explore a limited part of the global DEM (for instance, the model is cut in the middle of a dejection cone that is therefore difficult to identify).

Scale and orientation markers were not provided at the time but could be loaded in the 3D model directly via free software like 3D builder or Blender (see https://skfb.ly/onW7t for instance). Interactive tools were not used at that time, but some promising new functions are developed and could be used in the future (see https://labs.sketchfab.com/experiments/measurements/#! /models/3070ae00d83844e680ead63292140e43 for distance measurements on a 3D model, or V3Geo platform associated tools).

More importantly, as each student works on its own model, it is difficult to show structures and to guide them through the outcrop. Furthermore, there is no connection between the 3D model and the GIS project built in the first part of the lab session, therefore, no further mapping work could be done based on fieldwork measurements.



When the students are immersed in their own VR environment with guidance from outside the environment ("Alone in the field" or "single user" strategy), the user experience is greatly improved. Firstly, the students are facing the outcrop and can suddenly perceive its scale and overall aspect, they avoid common misinterpretations. The shared screen facility mentioned above is then absolutely necessary for the teacher to guide each student and provide them with personalized advice. Often spotted technical problems such as difficulties in using travel tools or compass on the plane can easily be detected and solved by a one-to-one discussion with the student. The discussion is highly facilitated between the teacher and each student since the teacher directly sees the zone the student is currently looking at and eventually pointing some specific details that can be orally discussed. The multiple shared screen facility is also of great help for students that get sick in VR since they can follow their colleague exploration on a screen and share their observations.

The most serious drawback of the "Alone in the field" strategy is the difficulty for the teacher to conduct some group briefings as it is commonly done during field class. In real fieldwork situations, after leaving the students looking at the outcrop for a while possibly with an exercise, briefings led by the teacher are times when group discussion and experience sharing take place between students, but also when the teacher can correct some misconceptions and show the students some key observations that could have been missed before. With an external person to the VR environment guidance and without sharing a common view of the outcrop with all the students, those briefings are mainly oral and too theoretical.

With this respect, the "Together in the field" or "multi-user" strategy is very close to real fieldwork teaching. Students and teachers are sharing the same virtual environment, and are viewing each other as avatars whose motions are visible (which gives a sensation of reality, see figure 5). Group briefings become then very natural and students can feel less isolated than in previous strategies. Some drawbacks obviously remain: first, as during real fieldwork and classes, students can be distracted and have the temptation to play with their colleague's avatar (which usually doesn't last long, see figure 5-c for instance); second, as during real fieldwork also, when students are exploring the outcrop on their own they can end up gathered in a specific place of the 3D model and disturb each other measurements; third, the motion of several avatars in the field of vision can increase the sickness for some users; and finally, because all microphones are shared via a video-conference facility, there is no room for private talks, chattering or whispering. This last aspect could be useful since even the comments in hushed voice of shy students could be clearly heard by the teacher.

The presence of a teacher in the VR environment that is physically abroad (as tested in our case between the classroom in Lyon and the teacher in Nice) opens novel perspectives for fieldwork. Indeed, students can benefit from guidance and advice from a researcher who is an expert of a given area or discipline and could therefore present in details how fieldwork observations are used in the current research process. Note that such work requires two teachers at the same time. A first expert is present in the VR environment. The second teacher is physically on site and can help student with controlers, or discuss with students who prefer being outside VR by directly showing structures on the shared screen.

Unfortunately, because of the sanitary restrictions, we ran out of time to conduct properly the last steps of the lab session. We initially wanted the students to make an observation diagram of the outcrop directly in their notebook, but this task requires time for going in and out the VR environment. We also wanted the students to load the measurements made in VR into their QGIS



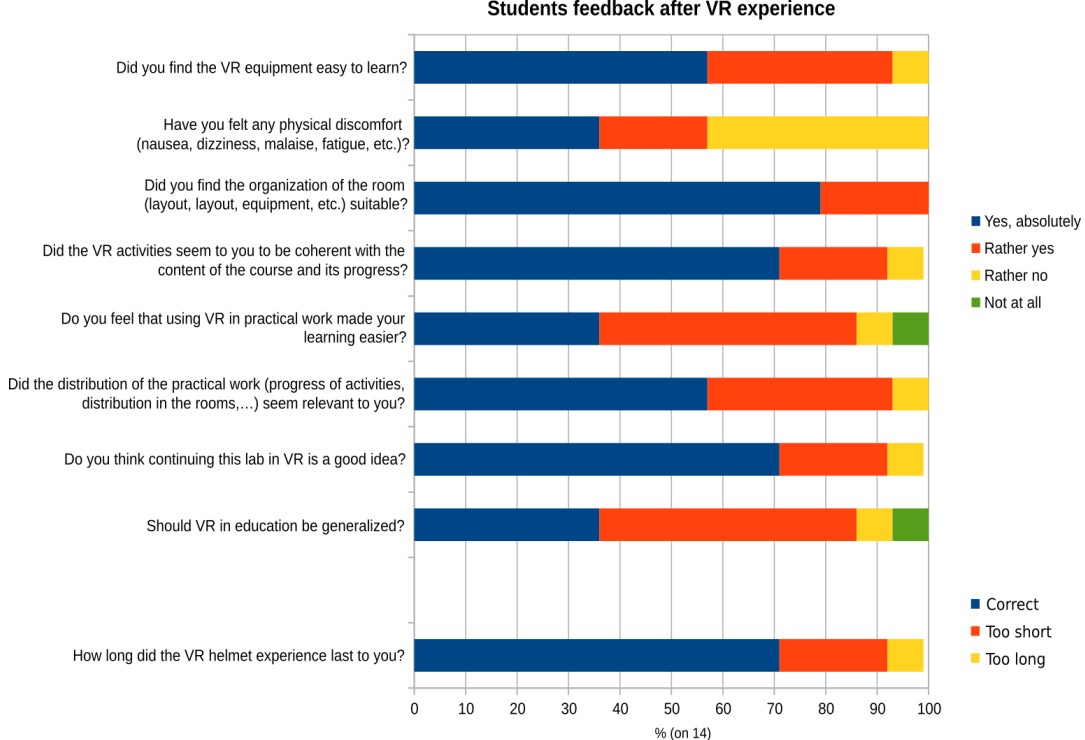

**Figure 6.** Results from the online inquiry proposed to the students who had the VR lab session. 14 out of 20 have answered.

project. This last step could be easily done in a standard 4h lab session and would probably help the students understanding the 3D to 2D relationship together with giving them the skills to properly map their own field measurements.

## 4.2 Learning in VR: students feedback

An online inquiry was carried out in the days following the lab-session to collect the students feedback on the VR experiment. 14 students answered (70% of the whole class). Even if not representative, the results presented in figure 6 give a first overview of their feeling after this unusual lab-session.

It is to note than 3 out of the 20 students that participated to the VR lab session in 2021 were not able to stay in the VR environment for more than few minutes because they were sick and felt uncomfortable. They were able to follow the session

by looking at the projected shared screens. More than 60% of the students who answered the survey say they have experienced a certain level of discomfort, that is not negligible. However, one student that expressed concern before trying the VR because she is subject to aquaphobia (abnormal fear of water and being immersed) was able to perform very well.

In general, no major difficulty arises from the controllers handling which is confirmed by the survey and was rather unexpected by the teaching team. The students felt that the virtual lab is adapted to the lab session and that the use of VR is





both consistent with the progress of the course and could help them learning. Logically, students that felt sick were not so enthusiastic.

Some students have left some detailed remarks about the VR lab-session that are listed below.

– *"It was really great, I loved the experience and I find that it opens new perspectives if it is done in complement to the real field trips. The duration of the immersion was a bit long, we were 4 out of 5 with nausea at the end, and I had sweating*
*that stung my eyes. Maybe the COVID mask also accentuated this effect."*

– *"It might be interesting to split the time with the VR headset with breaks to limit symptoms such as nausea, dizziness etc"*

– *" The only drawback is that it is difficult to take notes during the lab session. It must be done from memory afterwards."*

– *"Too bad not to use what we saw in the virtual lab afterwards."*

– *"Interesting even if for me nothing should replace the human contact between the teacher and the student, the VR can be a real BONUS. The course was very interesting but the expectations on the field were not so clear."*

These remarks clearly point two serious drawbacks of these first attempts to use VR in lab-sessions, that are first the physical discomfort that very often comes after an extended immersion and second the need for going back to 2D mapping after the virtual fieldwork to really integrate the observations into the tectonic analysis conducted with QGIS.

**4.3 Perspectives**

In general, the "Together in the field" or "multi-user" option appears to be the more promising and should be developed in the coming years. However, both the "100% virtual" and "Alone in the field" strategies could be improved and useful in the future depending on the intended use. If personal work is required, then it appears more appropriate to make the students work on a 3D model hosted directly on the web if interactive tools are available rather that asking them to install the Minerve software
and provide them with a VR apparatus, especially since the software requires an expensive computer with a powerful graphics card. To evaluate one student's skills at the end of the class, the "Alone of the field" configuration could be more adapted so that no interaction could be easily done with colleagues while the teacher could follow the behavior of the student on the field.

Some technical improvements could make the VR experience more efficient and will be considered for future development in the Minerve software. There is for instance a need for a laser pointer visible by all participants in the shared virtual environment,
that is for the moment not the case: each participant can see its own pointer. Being able to recognize the students and teachers avatar could help for personalized discussions (different colors or apparent names could be included). When working in a "Together in the field" configuration, each student can save its own measurements, but cannot share them with the other participants that could slow team work. Finally, the sound environment could be improved using spatialization technologies that provide perception of the sound depending on the distance. These techniques are often used in video games and VR and
could be implemented in Minerve (e.g. Tsingos et al., 2004, 2009).





These first lab-sessions using VR and the Minerve software have been conducted during time-limited sessions that imposed being nearly continuously immersed in the virtual environment. This continuous immersion has two severe drawbacks: it favors physical discomfort and prevent students to take notes or report observations in their notebooks. Software development will allow taking snapshots and notes, and in order to limit motion sickness, a better management of the teleportation could

be implemented as well as other techniques aiming at limiting this discomfort such as the reduction of the field of view during displacements (e.g. Fernandes and Feiner, 2016). Moreover, in future, longer sessions will be dedicated to this virtual fieldwork and regular pauses should be imposed to the students. This could be done by a detailed scripting of the course including exercises as schematic representations of some observations and reporting of measurements. In the future, students could also work by pairs, one student being immersed in the VR environment and the other guiding and taking notes. Roles in

the students pair should be exchanged during the session. Last but not least, significant time should be dedicated to uploading the measurements made on the field into the QGIS project and carrying out a final briefing.

## 5  Conclusions

Learning how to observe and interpret outcrops is one of the most important skills that Earth Sciences undergraduate students should learn, and field camps are ideal places to do so. They are however sometimes impossible to set up due to major

physical disabilities of the students, pandemic, or inaccessibility of the outcrop (high-altitude or deep submarine outcrops, active volcanoes, unique outcrop located very far from the teaching location, planetary bodies, etc). Virtual alternatives are being more and more considered by the Earth Sciences community via 3D online models or virtual reality tools, as evidenced by this special issue.

In this study, we have presented an attempt to take the third year undergraduate students of the *Observatoire des Sciences de*

*l'Univers de Lyon* to a very well preserved active fault scarp, a unusual fieldwork in metropolitan France. We take advantage of the very detailed bathymetric study conducted by the ODEMAR and SUBSAINTES french cruises in the Lesser Antilles over the Les Saintes plateau and the Roseau fault in particular. This submarine fault produced the 2004 Les Saintes Mw 6.3 normal fault earthquake and is exceptionally well preserved in the bathymetry.

In a first step, the students explore the seismo-tectonic context of the Les Saintes earthquake using georeferenced data

gathered on a QGIS project and results coming from scientific publications. Then, we use both the Sketchfab online utility and the Minerve open-source virtual reality software developed by Billant et al. (2019) to explore the detailed DEM/DOM of the fault scarp with the students. We chose three different configurations to conduct this virtual fieldwork that all present advantages and drawback that we analyse from teachers and students feedback. Even if our study is not representative given the small number of students involved, we find that the Minerve virtual reality software, when used in "multi-user" mode (i.e.

the students are sharing the same virtual environment together with the teachers), provide a very satisfactory framework that could still be technically improved. Students can measure strike, dip, rake, orientation and save their observations in a file that could be imported in any SIG software afterwards. Teachers find this mode very flexible. Interestingly, it is possible to combine various scales of observation from large landscape view to very fine observation on the outcrop and offer a good interactivity

with the students. Moreover, a specialist of the outcrop or of the thematic can be virtually present providing he/she is equipped
with headset, controlers and has the Minerve software locally installed. In the future, students skills should be evaluated in a
systematic way before and after the VR lab session to measure its teaching efficiency.

Students are in general enthusiastic to experiment virtual fieldwork even if physical discomfort is common. This could
be reduced by shortening the duration of the sessions in immersion in the virtual environment, ensuring breaks, and adding
software solutions. Even if it is clear that a virtual observation does not replace a field observation, Virtual Reality could be
a fantastic tool to bring students on remote or even inaccessible places such as submarine fault scarp or planetary bodies. It
therefore opens new perspectives for teaching Earth Sciences and we plan to use the Minerve software in other contexts in the
coming years.

*Code and data availability.* The Minerve software is developed under open-source license and is available on request. The centimetric
DEM/DOM is not open-source but can be available on request. Data used during the QGIS mapping exercise are available as supplementary
informations.

*Team list.* Nora Van Reeth, Félix Moquet and Amandine Minjollet

*Author contributions.* J.B developed the Minerve software. J.B, M.M and J-E.M designed the teaching experiment and actually taught. A.M
helped during the lab-session and the class conception. J.E and F.L gave access to the scientific data, provide an expert external view and
were beta testers for the Minerve software. The ICAP team took care of the technical issues and helped in the virtual lab use. All authors
contributed to the writing of this article.

*Competing interests.* The authors declare no competing interests.

*Acknowledgements.* The authors would like to thank all the researchers, technicians and crew involved in the ODEMAR and SUBSAINTES
cruises, whose findings are open for educational purposes. We thank C. Deplus and N. Feuillet for sharing the large scale 25m DEM from
Bathysaintes cruise. We thank Jùlia Bozzinio and Loïc Dornel for their participation in the development of Minerve. We are grateful to L3
students from 2019-2020 and 2020-2021 promotions for their feedback and resilience during this very difficult period. Special thanks are due
to Felix Moquet and Nora Van Reeth from the ICAP team (U. Lyon) for their patience in explaining VR to beginners, to A. Triantafyllou for
useful discussions and perspectives, and A.Minjollet for help with the figures. This project was partially supported by ANR Sersurf (ANR-
17-687 CE31-0020). We also thank the QGIS development team and we use base map and data from OpenStreetMap and OpenStreetMap
Foundation.



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
