# Peer review of "Deep oceanic submarine fieldwork with undergraduate students, an exceptional immersive experience (Minerve software)"

_Solid Earth, 2021_

## Referee Comment (RC1)

Review of Metois et al, "Deep oceanic submarine fieldwork with undergraduate students, an exceptional immersive experience (Minerve software)".

*General Overview*

This paper, by Metois *et al.*, describe a teaching experiment conducted at University of Lyon with a group of third year's undergraduate. It aims at providing a new VR experience to learn and experiment tectonics on a virtual field. The chosen example of a sub-marine outcrop is perfectly representing the challenge that VR will allow to overcome, such as the accessibility of the outcrop, which is situated abroad (in the Lesser Antilles in the Caribbean Sea) and underwater by ~1200m.

While the overall content would rather better fit a teaching-oriented journal like "Geoscience Communication" than "Solid Earth", it still falls well into the editorial line of the special issue. Specific examples of such VR fieldwork and/or remote teachings are still scarce (e.g., PlanMap planetary mapping winter school), and this work is a very welcome sight to support the ongoing effort to develop new way of teaching and diffusing geosciences, including the possibility presented by the authors to involve a guest specialist that is not physically present.

The author specifically present the dataset, their origin and the foreseen use as a mapping and tectonics interpretation exercise. They also present three different cases in which the students are 100% remote (using web-based resources, particularly representative of the Covid situation), and two *in situ* VR experiences, with and without a teacher. It comes out that student are quite supportive of the VR-based teachings, providing they are a supplement to "real in person" classes, which is expected from other experiments of the same kind. Nevertheless, it provides and proves that VR brings a strong added value when it comes to learning Geosciences.

Overall, the manuscript is well written and provides sufficient elements to get a good idea of the outcome of this experiment and invite us to participate in such efforts. I would therefore strongly recommend publishing this article in this form, providing minor corrections in the wording (see following).

*Specific comments*

The overall intent of the experiment, while rushed as explained by the authors because of the Covid pandemic breakout, relies on a strong premise and allows to get a solid understanding of what to expect from such practice. The choice of the object was also pertinent.

Initiation to controls and having the student look at the artifacts to prevent misinterpretation is a very good point.

As one feedback points out, maybe the social distancing and mandatory wear of a facial mask were also setbacks that prevent the student to be completely comfortable. This assertion should figure in the conclusive lines of 4.2 as the Covid situation is mentioned and should be more considered regarding this experience.

*Technical corrections; Detailed comments on the manuscript*

Figures:
- Figure 2 is a little small, and the color choice doesn't help getting a good idea of what is illustrated. I recommend using more contrasted colors.

Corpus of the manuscript:
- Line 40: Describe "AUV" acronym
- Line 53: inverse "inspect" and "interactively" words (adverbs come first)
- Line 59: add "degree" after "License"
- Line 63: replace "slow" by "low"
- Line 70: Please keep consistency. You wrote in Line 59 ("3$^{rd}$" and here "third", as well as "undergraduate", or "license". Please keep the same terminology, possible the undergraduate third year so that foreign readers could understand which students you are talking about
- Line 76: Timeline is written as a single word without hyphen
- Line 200: Prefer "students that suffer motion-sickness induced by VR" to "student that get sick in VR" as it relates the actual condition
- Line 201: Replace "colleague" by "classmate" (students are not colleagues)
- Line 206: Rephrase "With an […] guidance" for clarity
- Line 262: Replace "colleague" by "classmate"

---

## Author Response (AR1)

We would like to thank both reviewers for their useful suggestions that we have taken into account where possible. We hope this new version of the manuscript is clearer, better sourced and that the supplementary materials are now practically usable by colleagues, which is of the most importance to us (companion files to .shp were missing in the previous archive). We also propose to add a supplementary video showing an extract of the interactions possible in the Together in the field strategy.
Please find in the following detailed answer to the reviewer's comments.

**############## Reviewer 1 – Gwénaël Caravaca**

**General Overview**
This paper, by Metois et al., describe a teaching experiment conducted at University of Lyon with a group of third year's undergraduate. It aims at providing a new VR experience to learn and experiment tectonics on a virtual field. The chosen example of a sub-marine outcrop is perfectly representing the challenge that VR will allow to overcome, such as the accessibility of the outcrop, which is situated abroad (in the Lesser Antilles in the Caribbean Sea) and underwater by ~1200m.
While the overall content would rather better fit a teaching-oriented journal like "Geoscience Communication" than "Solid Earth", it still falls well into the editorial line of the special issue.

We agree with the scope of the journal, but the special issue is common between SE and GC so we hope the paper would reach a broad audience in the teaching community.

Specific examples of such VR fieldwork and/or remote teachings are still scarce (e.g., PlanMap planetary mapping winter school), and this work is a very welcome sight to support the ongoing effort to develop new way of teaching and diffusing geosciences, including the possibility presented by the authors to involve a guest specialist that is not physically present. The author specifically present the dataset, their origin and the foreseen use as a mapping and tectonics interpretation exercise. They also present three different cases in which the students are 100% remote (using web-based resources, particularly representative of the Covid situation), and two in situ VR experiences, with and without a teacher. It comes out that student are quite supportive of the VR-based teachings, providing they are a supplement to "real in person" classes, which is expected from other experiments of the same kind.
Nevertheless, it provides and proves that VR brings a strong added value when it comes to learning Geosciences.
Overall, the manuscript is well written and provides sufficient elements to get a good idea of the outcome of this experiment and invite us to participate in such efforts. I would therefore strongly recommend publishing this article in this form, providing minor corrections in the wording (see following).

We thank the reviewer for its support and acknowledgment of our efforts.

**Specific comments**
The overall intent of the experiment, while rushed as explained by the authors because of the Covid pandemic breakout, relies on a strong premise and allows to get a solid understanding of what to expect from such practice. The choice of the object was also pertinent.
Initiation to controls and having the student look at the artifacts to prevent misinterpretation is a very good point.
As one feedback points out, maybe the social distancing and mandatory wear of a facial mask were also setbacks that prevent the student to be completely comfortable. This assertion

should figure in the conclusive lines of 4.2 as the Covid situation is mentioned and should be more considered regarding this experience.

Agreed. We now state at the end of 4.2 this specific context : "We also must recognize that the VR lab-session took place in a very specific context, i.e. the situation of partial restrictions in teaching activities due to Covid-19 pandemics in spring 2021 in France. This overall context has imposed the partial use of facial masks in the virtual lab and physical distancing, which could influence the student's confort during the VR experiment."

**Technical corrections; Detailed comments on the manuscript**

All technical corrections have been addressed in the new version of the manuscript to be submitted.

Figures:

- Figure 2 is a little small, and the color choice doesn't help getting a good idea of what is illustrated. I recommend using more contrasted colors.

Corpus of the manuscript:

- Line 40: Describe "AUV" acronym - This acronym is now spelled out
- Line 53: inverse "inspect" and "interactively" words (adverbs come first)
- Line 59: add "degree" after "License"
- Line 63: replace "slow" by "low"
- Line 70: Please keep consistency. You wrote in Line 59 ("3 rd " and here "third", as well as "undergraduate", or "license". Please keep the same terminology, possible the undergraduate third year so that foreign readers could understand which students you are talking about
- Line 76: Timeline is written as a single word without hyphen
- Line 200: Prefer "students that suffer motion-sickness induced by VR" to "student that get sick in VR" as it relates the actual condition
- Line 201: Replace "colleague" by "classmate" (students are not colleagues)
- Line 206: Rephrase "With an [...] guidance" for clarity
- Line 262: Replace "colleague" by "classmate"

**Reviewer 2**

**General Comments**

This manuscript presents virtual field exercises to teach practical skills to students as an alternative to 'long and expensive field-work sessions'. The authors describe several delivery methods, including one where the authors used a dedicated VR room and software that allows research collaboration in VR (an experiment described in a cited conference paper). The methods described allowed students to observe, explore and experience an otherwise inaccessible, exceptionally preserved surface fault rupture outcrop through virtual reality using a digital model that was created for research purposes, and use their observations to learn about earth processes.

This manuscript impressed me for several reasons. The authors describe the decisions and reasoning, as well as the learning that they were trying to achieve, when designing their virtual experience. They list three formats of delivery, and comment on the benefits and drawbacks of each method, all while acknowledging their own shortcomings. The authors provide student feedback on their experience whilst learning and engaging with the content. And finally, the authors have chosen software that is free and open source, including the software that they developed themselves, as well as making all of their learning resources available through this special issue. This latter point means that it would be extremely simple to build on or attempt to replicate the author's experience.

My impression is that the authors present a balanced and honest view of their experiment of teaching with VR. At the moment, the quality of the paper does suffer from quite a few language and grammar errors that should be easily rectified.

We carefully reviewed the paper grammar and took into account the many phrasing comments of the reviewer that we thank for his/her time in pointing them out.

The paper might also benefit from a clearer collection methodology on the data used (the teacher reflection and student feedback).

We add a paragraph at the beginning of section 4.1 to detail the teacher's profile and discussions set up in the team (l.182).

The discussion on the data could also use some grounding in relevant literature.

Yes indeed. We recognize that we are lacking a deep knowledge of the literature relative to teaching strategies. In the new version of the manuscript, we took advantage of the reviewer's suggestions and now propose a broader bibliography.

The paper does not show (nor attempt to) any clear improvement on learning outcomes; however, it does highlight the enthusiasm that student had engaging with the content.

Fair enough. Because of the limited duration of the experiment and of the covid-19 constraint we were not able to conduct proper assessments that would have enabled us to compare the learning outcomes of students that have or have not participated in the virtual field work. Expected outcomes significantly differ from one teaching strategy from another (for instance consistent dip measurements cannot be expected in the 100% virtual mode) that would make this comparison difficult. However, in the coming years, we will include assessments during and after the virtual-lab session. We now discuss this point at line 288.

Therefore, I believe readers will be encouraged to follow in the authors footsteps, use the authors resources, and experiment with VR in the classroom. I believe that after a review of the comments, the manuscript will form an excellent complement to the Solid Earth special issue on virtual field experiences.

We thank reviewer 2 for his/her comments and encouragement. Follow a point-to-point answer to more specific comments.

**Specific comments**

First, as highlighted before, the paper makes exclusive use of free and open-source software (FOSS), removing one major hurdle for colleagues wishing to replicate this experiment in their own setting. The other benefit of the use of FOSS is that students can easily take the skills learnt during this lab with them into their future careers once they leave university, without having to retrain using new software. I believe the authors could highlight this positive aspect of their work.
This is indeed an important point for all of us. Developing FOSS is a long-term investment that will hopefully be useful for a broader community. We emphasize this point in the new version of the manuscript (l.59, l.86, l.98).

The authors have taken extra effort to provide to students with carefully curated course material, digital files and software settings (e.g., L96-98) to ensure little to no technical difficulties. Can the authors specify if plugins are required to be installed separately by students(L105), and if yes are students given guidance on how to install plugins?

The only required additional plugin is the Terrain Profile tool that is included in the standard Qgis Official Plugin repository. Therefore, activating the plugin is very simple and requires you to tick the install box in the "Manage and Install Plugins" window. We add this step in the 3.1 section describing the use of QGIS (l.115).

Are students given guidance on how to use the layout manager (L110), and if they are, how?
Are students shown how to import a csv file into their project (L158-160)?

We add these details in the new version (l.110) :
"Guidance on this first use of SIG software is provided via online discussions on the Discord app forum and teachers-made video tutorials in the 100% remote strategy while oral explanations are provided in the classroom for the two other strategies."
We could not go to the csv import step due to lack of time, but the format exported from Minerve can directly be imported as csv in QGIS without additional tuning.

One theme throughout the paper is the replication of 'real fieldwork' (L171, L203, L208). Are the authors really trying to approximate 'real fieldwork' with the VR experiment? I would encourage the authors to shift their thinking when it comes to virtual field experience from replicating real field work, to replicating learning outcomes that occur in the field. Some learning outcomes might prove to be achievable in virtual field experience (in fact some learning outcomes might be better in virtual experiences!), however some might not. E.g the authors highlight how this experience might have helped solve some accessibility issues for one student, who could never have gone to field if they wanted to (L236) due to a phobia ( if this was possible..). I think it is important to think of virtual field experiences as one tool to achieve learning outcomes instead of as a replacement for in person field trips.

We agree with this comment since teaching during virtual fieldwork cannot be a replication of classical on the field teaching. We try in the new version to avoid the "real fieldwork" expression and to highlight the outcomes made available via the virtual experience (GIS mapping in particular) in the discussion and conclusion sections.

One particularly exciting feature of the virtual field method together in the field method is the ability of sharing the field experience with others, as we know that peer feedback helps increase understanding of concepts see refs.
Falchikov, N. (2001). Learning together: Peer tutoring in higher education. Psychology Press.,or Duret, D., Christley, R., Denny, P., & Senior, A. (2018). Collaborative learning with PeerWise. Research in Learning Technology, 26, 1-13.)

Agreed, we read and included these references when arguing for the relevance of sharing and collaborating during VR experiments.

Another general suggestion would be that the manuscript would benefit from referring to relevant literature. For example in section 4.3 perspectives, the authors make a lot of excellent suggestions on improvements to their exercise, but their arguments would be much more convincing if there was literature that supports those suggestions. Some examples from that section would be: Why do it be an improvement to recognise avatars in VR? Why is slow teamwork a negative effect? These are just two small examples, however I would encourage the authors to tie all of their discussion back to relevant literature.

We reckon that our knowledge of the relevant literature is incomplete. In the new version of the manuscript we include some of the reviewer2's suggestions and added others.

L174 Was the participation in oral discussions part of the course assessment? How was their understanding of concept assessed in the remote strategy? In general, was there any assessment associated with this exercise? The assessment strategy might be of interest to readers.

See answer to general comment. We added this paragraph at the end of the discussion section (l.284) :
"Finally, this experiment does not allow us to definitively conclude on the efficiency of our strategies in student's learning because learning outcomes were not assessed in the ``Alone in the field'' and ``Together in the field'' strategies conducted in 2021. Theoretical active tectonics related skills were tested in 2020 following the ``100\% remote'' lab session and related course via online assessments during which students had to analyze maps presenting fault traces, focal mechanism, coseismic surface displacements and had to make first order calculations based on usual scaling laws. In future, we would like to evaluate the understanding of the Les Saintes virtual fieldwork by asking for both an interpretative structural map of the area built on QGIS and an observational scheme of the DOM."

L175 What type of issues are required to be troubleshooted? Are the issues related to the use of software, to remote working, to the innovative nature of this exercise?
We add details in this section on which problems have been faced by the students when using QGIS (l.188) :
"Lab sessions in-person are more efficient to help the students both with technical issues (QGIS is already installed in the latest version, plugins can be easily installed, problems in saving the new shapefiles can be directly solved, etc) and for guiding in the interpretation of the DEM (by ensuring the student is looking at the proper structure)"

L195 "they avoid common misinterpretations." What common misinterpretations?
We precise (l.211) :

Firstly, the students are facing the outcrop and can suddenly perceive its scale and overall aspect, they avoid common misinterpretations that can be due to localized artifacts in the DEM for instance or because no common scaling is visible at first glance on the outcrop since we are underwater (no flora or fauna elements)

L224 – Did you run out of time for the "together in the field" version only? Is there anything that can be done to use your time more efficiently, or is the time problem inherent in using VR as a teaching tool?

We added some details on this point (l.246) : "Unfortunately, because of the sanitary restrictions that have imposed to split the students in smaller groups and to divide the dedicated sessions in 2h sessions rather than 4h, we ran out of time to properly conduct the last steps of the lab session for both the ``Together'' and ``Alone in the field'' strategies."

We are confident that, when sanitary restrictions are over and we can gather 12 students at the same time in the virtual lab in 4h lab-sessions we will be able to conduct the full program without hurry.

L229. Instead of saying "would probably help the student understanding", it might be more appropriate to say that "4 th lab session could be designed around synthesising 3D and 2D data .." , the difference being that the latter is a learning design, and the former a prediction of learning achieved by students..

Ok, rephrased.

L242 are the quotes translated? It would be worth specifying that, also as there are some grammatical errors in the quotes that might have been introduced by the authors.

Yes this is a translation from french (that sometimes contains grammatical errors…), this is now stated in the text.

L298 What do you mean "not representative"? Not representative of a wider population of users' experience? I don't think it is necessary to mention the population size when concluding on the success of this experiment, I would just say that it worked well for your group. That means that you are not trying to generalise your results to a wider group.

Fair enough, we rephrased. Based on the students and teachers feedback, we find that the Minerve virtual reality software, when used in ``multi-user'' mode (i.e. the students are sharing the same virtual environment together with the teachers), provides a very satisfactory framework that could still be technically improved.

We have taken in consideration all the technical suggestions from Reviewer2 and we thank him/her again for this careful reading.